# Perilesional Targeted Biopsy Combined with MRI-TRUS Image Fusion-Guided Targeted Prostate Biopsy: An Analysis According to PI-RADS Scores

**DOI:** 10.3390/diagnostics13152608

**Published:** 2023-08-07

**Authors:** Masayuki Tomioka, Kensaku Seike, Hiromi Uno, Nami Asano, Haruo Watanabe, Risa Tomioka-Inagawa, Makoto Kawase, Daiki Kato, Manabu Takai, Koji Iinuma, Yuki Tobisawa, Keita Nakane, Kunihiro Tsuchiya, Takayasu Ito, Takuya Koie

**Affiliations:** 1Department of Urology, Gifu University Graduate School of Medicine, Gifu 5011194, Japan; ch.a3200@gmail.com (M.T.); s11110032@gmail.com (R.T.-I.); kawase.makoto.g5@f.gifu-u.ac.jp (M.K.); kato.daiki.m2@f.gifu-u.ac.jp (D.K.); takai.manabu.a5@f.gifu-u.ac.jp (M.T.); iinuma.koji.s0@f.gifu-u.ac.jp (K.I.); tobisawa.yuki.a7@f.gifu-u.ac.jp (Y.T.); nakane.keita.k2@f.gifu-u.ac.jp (K.N.); 2Department of Urology, Chuno Kousei Hospital, 5-1 Wakakusadori, Seki 5013802, Japan; k-seike@chuno.gfkosei.or.jp (K.S.); h-uno@chuno.gfkosei.or.jp (H.U.); 3Department of Pathology, Chuno Kousei Hospital, 5-1 Wakakusadori, Seki 5013802, Japan; nasano@chuno.gfkosei.or.jp; 4Department of Radiology, Chuno Kousei Hospital, 5-1 Wakakusadori, Seki 5013802, Japan; haruwow@live.jp; 5IROHA Home-Care Clinic, Ogaki 5030034, Japan; ktsuchiya@iroha-care.com; 6Center for Clinical Training and Career Development, Gifu University Graduate School of Medicine, Gifu 5011194, Japan; ito.takayasu.v9@f.gifu-u.ac.jp

**Keywords:** prostate cancer, prostate biopsy, targeted biopsy, perilesional targeted biopsy, grade group, Prostate Imaging Reporting and Data System score

## Abstract

A prostate-targeted biopsy (TB) core is usually collected from a site where magnetic resonance imaging (MRI) indicates possible cancer. However, the extent of the lesion is difficult to accurately predict using MRI or TB alone. Therefore, we performed several biopsies around the TB site (perilesional [p] TB) and analyzed the association between the positive cores obtained using TB and pTB and the Prostate Imaging Reporting and Data System (PI-RADS) scores. This retrospective study included patients who underwent prostate biopsies. The extent of pTB was defined as the area within 10 mm of a TB site. A total of 162 eligible patients were enrolled. Prostate cancer (PCa) was diagnosed in 75.2% of patients undergoing TB, with a positivity rate of 50.7% for a PI-RADS score of 3, 95.8% for a PI-RADS score of 4, and 100% for a PI-RADS score of 5. Patients diagnosed with PCa according to both TB and pTB had significantly higher positivity rates for PI-RADS scores of 4 and 5 than for a PI-RADS score of 3 (*p* < 0.0001 and *p* = 0.0009, respectively). Additional pTB may be performed in patients with PI-RADS ≥ 4 regions of interest for assessing PCa malignancy.

## 1. Introduction

Magnetic resonance imaging (MRI)-guided targeted biopsy (TB) of the prostate involves minimally targeted sampling of the region of interest (ROI) identified using MRI [1]. Compared to ultrasound-guided systematic prostate biopsy (PBx), this method has been shown to detect a higher proportion of clinically significant prostate cancer (csPCa) and a lower incidence of insignificant prostate cancer (PCa), and it may reduce complications associated with PBx [1,2,3]. Therefore, several guidelines recommend that a combination of systemic PBx and TB be performed [4,5]. Indeed, the positive detection rate of PCa using a combination of TB and systematic PBx was significantly higher than that using TB or systematic PBx alone in our previous studies [2,3,6]. However, the actual size of PCa was also reported to be larger than the ROI identified using MRI [7,8,9]. Among patients with csPCa, a significant proportion had cancer within the ROI, while others had cancer located outside the ROI [7]. Furthermore, there is a notable correlation between the extent of csPCa and the obtained PI-RADS score when defining the radius centered on an ROI where cancer is easily identified as the “penumbra.” For example, at a PI-RADS score of 3, the width of the 90% penumbra was approximately 16 mm, whereas at a PI-RADS score of 5, the width was only 5–6 mm [7]. Still, the precise distribution of PCa in ROIs and the extent to which systematic PBx can improve sensitivity with respect to cancer detection rates remain controversial [7]. These reasons include the fact that MRI may not detect PCa, tumor heterogeneity, and various targeting errors in the fusion device used to perform MRI-transrectal ultrasound (TRUS) image fusion TB [7]. Additionally, some cores collected in systematic PBx were reported to be obtained from areas close to an ROI, as pointed out on MRI (perilesional) [10]. Similarly, csPCa was found to be present in the perilesional core in retrospective reviews of systematic PBx [11,12]. In a consensus expert opinion of the Prostate Imaging Reporting and Data System (PI-RADS) Committee, perilesional sampling was found to be useful in detecting PCa [8,13]. However, the true utility of perilesional (p) TB and a clear definition of the extent to which biopsy cores should be collected remain unclear [12]. Furthermore, there have been few reports on the correlation between PI-RADS scores and the positivity rate of TB around ROIs or on whether PI-RADS scores affect the difference in the detection rate of csPCa between an ROI and its surrounding sites [7,12,13].

In the present study, we analyzed the PCa detection rate and grade group (GG) discrepancy between TB and pTB in patients with a PI-RADS score ≥ 3 and examined the significance of performing TB in the area surrounding an ROI.

## 2. Materials and Methods

### 2.1. Patient Population

The Institutional Review Board of the Chuno Health Hospital approved this study (Approval No.: R4-8). As this was a retrospective study, informed consent from the patients was exempted. In addition, written consent was deemed unnecessary in accordance with the provisions of the Ethics Committee and Ethics Guidelines in Japan. This was based on the fact that the results of a retrospective and observational study utilizing existing materials had already been published.

This retrospective study was conducted on patients who underwent transperineal MRI-TRUS image fusion-guided TB using the KOELIS Trinity^®^ system between September 2020 and December 2022 at the Chuno Kosei Hospital. The indication for biopsy using this method was a PI-RADS v2.1 score of ≥3 on MRI prior to biopsy. Peripheral TB was defined as a biopsy of the area within 10 mm of an ROI as noted using MRI. Patients with multiple ROIs on their MRI were excluded from the study.

### 2.2. Multiparametric Magnetic Resonance Imaging (mpMRI)

All patients underwent mpMRI using a MAGNETOM Skyra 3 Tesla system (Siemens Healthcare, Erlangen, Germany) with an 18-channel coil. The mpMRI images were evaluated by one radiologist (H.W.) and one urologist (Mas.To.), each with more than 10 years of clinical experience based on the PI-RADS v2.1 criteria. The PI-RADS scores was determined through discussion between the two physicians mentioned above to make the final decision.

### 2.3. Prostate Biopsy Protocol

TB of the prostate using MRI-TRUS image fusion was performed under spinal anesthesia using an 18G automatic biopsy gun. Depending on the size of the ROI, 2–4 cores were collected during the TB and 1–6 cores during the pTB. PBx was performed transperineally, and systemic PBx was performed in all patients.

### 2.4. Pathological Analysis

An independent pathologist (N.A.) at the Chuno Kosei Hospital evaluated the biopsy specimens according to the 2014 International Society of Urological Pathology criteria [14]. csPCa was determined as having at least one core of GG ≥ 2. Clinically insignificant PCa was defined as having a core of GG = 1.

### 2.5. Statistical Analysis

The primary endpoint of the study was the proportion of patients in whom PCa was detected at both the ROI and perilesional sites. The secondary endpoint was the proportion of patients with a higher GG in the biopsy specimens obtained using pTB compared to those obtained using TB. Pearson’s chi-square test was used to statistically compare different PI-RADS scores. The datasets for this retrospective study were analyzed using JMP Pro version 16.2.0 (SAS Institute Inc., Cary, NC, USA). Statistical significance was set at a two-tailed *p*-value < 0.05.

## 3. Results

Patients

Of the 172 patients who underwent MRI-TRUS image fusion-guided TB, 162 eligible patients were enrolled in the study (Figure 1).

All patients underwent PBx with TB using the MRI-TRUS image fusion target method with KOELIS Trinity (Koelis, Grenoble, France). In addition, pTB specimens were collected within 10 mm of the ROI using three-dimensional mapping (Figure 2). The number of specimens collected for biopsy was determined by the surgeon based on the size of the ROI.

A summary of the participants’ demographic characteristics is presented in Table 1. In all patients, PBx was performed via a transperineal approach. Prostate volume (PV) was measured using the KOELIS Trinity^®^ system. The median age, prostate-specific antigen level, and PV of the enrolled patients were 74 years, 6.8 ng/mL, and 36.5 mL, respectively. The median number of samples collected for TB and pTB was two, and the median number of samples collected for systemic PBx was 10. PCa was detected in approximately two-thirds of all patients, 63% of whom were diagnosed with csPCa. With regard to the length of cancer detected in the prostate biopsy specimens, the median length of cancer was 7 mm (interquartile range [IQR], 4–9 mm) in the TB specimens and 5 mm (IQR, 2–7 mm) in the pTB specimens, showing a significantly longer cancer length for TB (*p* = 0.021).

Figure 3 shows the distribution of patients in the PI-RADS score 3, 4, and 5 groups, classified according to the TB, pTB, and systemic biopsy test results. Histologically, 89 cases were diagnosed as positive for both TB and pTB, 9 were diagnosed as positive only for systematic biopsy, 20 were diagnosed as positive for TB and negative for pTB, and 5 were diagnosed as negative for TB and positive for pTB. Focusing on the group that was positive for both TB and pTB, the detection rate was significantly higher in the PI-RADS score 4 and 5 groups than in the PI-RADS score 3 group, which was 75.0% and 76.9%, respectively, compared to 28.6% in the PI-RADS score 3 group. (*p* < 0.001 and *p* < 0.001, respectively).

Figure 4 shows the association between the GG based on TB and that based on pTB in 86 patients in whom PCa was detected in both TB and pTB. For these 86 patients, a GG upgrade was observed in 29.1% of patients based on pTB compared to TB. The percentage of patients with GG upgraded by pTB over TB was 18.2% in the PI-RADS score 3 group, 35.2% in the PI-RADS score 4 group, and 20.0% in the PI-RADS score 5 group, with no statistically significant differences among the three groups.

On the other hand, among the patients classified as “no,” there were 11 cases in the PI-RADS score 3 group whose GG matched between their TB and pTB, while 7 cases showed downgrading in their pTB compared to TB. In the PI-RADS score 4 group, 13 of 35 patients had downgraded GG in their pTB compared to TB, while none of the patients with PI-RADS score 5 had downgraded GG in their pTB.

The “no” category contains cases of GG downgrade according to pTB. Of the 18 “no” cases in the PI-RADS score 3 group, TB and pTB showed the same GG in 11 cases and downgrade was seen in 7 cases. Similarly, in the PI-RADS score 4 group, 13 of 35 cases were downgraded, and in the PI-RADS score 5 group, 0 of 8 cases was downgraded.

## 4. Discussion

Localized PCa is a malignant neoplasm for which different treatment options can be selected, including active surveillance, surgery, and radiation therapy, depending on the risk classification [4,15,16]. Although it is crucial to identify the location and T stage of PCa and its grading by GG using MRI and PBx before treatment, it is currently difficult to diagnose the exact location of the tumor, as it is well recognized that PCa is a multifocal disease [16,17,18]. Among patients with clinical T1c PCa who underwent radical prostatectomy, 76% had multifocal disease [17]. To the best of our knowledge, this is the first study to examine GG discrepancies between TB and pTB. Additionally, we examined the correlation between PI-RADS scores and the rate of cancer positivity in TB and pTB. In the present study, we found that 29.1% of the pTB specimens had a GG upgrade compared to the TB specimens from patients with cancer that was detected in both TB and pTB. The percentage of cases with cancer positivity in both TB and pTB was significantly higher among patients with PI-RADS scores of 4 and 5 than those with a PI-RADS score of 3. These results may indicate a higher actual tumor volume in patients with PCa who had a higher PI-RADS score, and pTB may help to solve these problems. However, it is difficult to accurately determine the size, extent, and distribution of tumors prior to treatment using current examination methods.

Additionally, the PBx specimens showed a significantly greater cancer length and tumor volume in the multifocal PCa group than in the unifocal PCa group (both *p* < 0.05) [17]. In 486 patients who underwent radical prostatectomy, the mean volume of the largest tumor was 4.16 mL, while an average of 2.92 separate tumors were detected, with a mean tumor volume of 0.63 mL [18]. In our previous study, 372 patients underwent histopathological diagnosis and tumor volume assessment of prostatectomy specimens [19]. The mean tumor volume was 2.29 mL for the enrolled patients with a median age of 68 years, and the initial prostate-specific antigen level of 7.50 ng/mL [19]. Among them, 16.4% of the patients had a tumor volume <0.5 mL and insignificant PCa was found in only 3.7% of the cases [19]. In a meta-analysis comparing the detection rate of PCa using different methods of PBx with MRI, TB + pTB and TB + systematic PBx were compared, and the mean number of cores sampled was 9.5 for TBx + pTB and 16.5 for TBx + systematic PBx, with the latter having significantly more cores (*p* = 0.003) [9]. However, there was no significant difference in the detection rate of csPCa with GG ≥ 2 between the two groups (*p* = 0.09) [9]. Similarly, the detection rate of csPCa using TBx + pTB was not significantly associated with the total number of biopsy cores (*p* = 0.92) [9]. In contrast, TBx + pTB resulted in significantly more biopsy cores (*p* = 0.005) and a significantly higher detection rate of csPCa (*p* < 0.001) compared to TB alone [9]. Therefore, we believe that MRI and PBx should be carefully used to determine the treatment strategy because not all PCa cases can be detected by MRI and PBx.

For patients with a PI-RADS score ≥ 3 on their MRI, several investigators have suggested that only an MRI-based TB is sufficient and that a systematic PBx is not necessary [1,20]. Ahmed et al. [20] investigated whether mpMRI could identify the presence of csPCa based on template prostate mapping biopsy (TPM-biopsy) [20]. In this study, csPCa was defined as GG ≥ 3 or maximum length of the cancer core ≥ 6 mm [20]. A total of 576 patients underwent both TRUS-systematic PBx and TPM-PBx biopsy methods after mpMRI, with TPM-PBx showing PCa in 71% of the patients, 40% of whom had csPCa [17]. For csPCa, mpMRI was significantly more sensitive and less specific than TRUS-PBx (*p* < 0.0001 and *p* < 0.0001, respectively) [20]. Forty-four patients (5.9%) had serious adverse events caused by PBx, including 8 patients with urinary tract infection followed by sepsis [20]. It was suggested that mpMRI could avoid primary biopsy in 27% of patients and could reduce the diagnosis of clinically insignificant PCa by 5% [20]. Therefore, the authors concluded that mpMRI has the potential to reduce overdiagnosis of insignificant PCa and improve the detection rate of csPCa [20]. In contrast, Hugosson et al. [21] suggested that although there is a 50% chance of not detecting insignificant PCa if systematic biopsy is omitted, it is possible to miss high-grade PCa in 20% of cases. Although the detection rate of csPCa was not significantly different between systematic PBx and TB (*p* = 0.38), it was reported that csPCa could have been missed in 5.2% of patients without systematic PBx and in 7.6% of those without TB [22]. Therefore, the authors concluded that the combination of both methods could improve the detection rate of csPCa above the GG of 2 [23]. In our previous study, although the detection rate of insignificant PCa was significantly higher with systematic PBx alone than with TB (*p* < 0.001), the combination of TB and systematic PBx significantly increased the positivity rate of csPCa compared with that in patients who underwent TB or systematic PBx alone (*p* = 0.001 and *p* < 0.001, respectively) [2,3]. In our previous study examining the association between PI-RADS scores and PCa detection rate via mpMRI, 76.7% of patients with a PI-RADS score of ≥3 were diagnosed with PCa [3]. The PCa detection rate for patients with PI-RADS scores of 4 or 5 was significantly higher than for those with a PI-RADS score of 3 (3 versus 4, *p* < 0.001; 3 versus 5, *p* < 0.001; and 4 versus 5, *p* = 0.073) [3]. Of these patients, 113 (37.7%) were diagnosed with TB alone and PCa, while 43 (14.3%) had undetectable PCa [3]. In contrast, the detection rate of csPCa in patients who underwent TB + systematic PBx was 46% [3]. However, previous studies on the clinical usefulness of pTB are limited [24,25]. In a study of 294 patients with suspected PCa using mpMRI, overall, 150 cancers and 86 cases with GG ≥2 were diagnosed [25]. Systematic PBx missed 18 tumors with GG ≥2 (20.9%), while 11 (12.8%) were undetected based on TB [25]. Although the detection rates of PCa with GG ≥ 2 for both methods were not statistically different, primary TB showed a trend of being superior compared to systemic PBx (*p* = 0.08) [25]. To diagnose one PCa with GG ≥ 2, 3.4 PCa were needed for TB and 7.4 for systemic PBx [25]. Limiting biopsies to cases with PI-RADS scores of 3–5 would miss 19.8% of PCa above GG of 2, indicating the limited sensitivity of mpMRI [25]. Therefore, the authors concluded that a combination of TB and systemic PBx should be performed for PCa detection in primary biopsies [26]. In addition to TB, pTB of an area of approximately 5 mm around ROIs identified PCa in 24% of patients [10]. Additionally, csPCa with GG of 2–5 was detected by pTB in 9% of the patients diagnosed with insignificant PCa or no cancer by TB (*p* = 0. 004) [10]. When only lesions with PI-RADS scores of 4 and 5 were considered, TB of the ROIs detected csPCa in 70.8% of the cases; however, pTB yielded a diagnosis of csPCa in 12.6% of the cases when TB identified insignificant PCa or no cancer [10]. These results suggest the validity of performing an additional biopsy of the penumbral region for ROIs with PI-RADS scores 4 and 5 [10]. Regarding TB for an ROI, the median GG was 2 and the longest tumor length was 7.0 mm, whereas the median number of biopsies near the target was four, with two positive cores; the median highest GG was 2, and the median longest tumor length was 6.5 mm [22]. Utilizing both TB and pTB may increase the accuracy of csPCa detection [22]. Outside the ROI, the csPCa detection rate was 30%. The detection rate of csPCa according to PI-RADS scores was 8% for lesions with a PI-RADS score 2, 15% for lesions with a PI-RADS score 3, 36% for lesions with a PI-RADS score 4, and 58% for lesions with a PI-RADS score 5 (*p* = 0.03) [27]. In the multivariate analysis, PI-RADS scores of 4 and 5 were also independent predictors of csPCa outside the ROI [27].

There have been several reports of discrepancies between the GG of prostate biopsy specimens and that of specimens removed after radical prostatectomy [27,28,29,30,31]. A comparison of PBx and prostatectomy specimens was reported with respect to GG and showed that 20.5% of surgical specimens had upgraded GG and 18.1% had downgraded GG [29]. The addition of TB resulted in an upgrade of GG in 32%, while systematic PBx detected GG ≥ 3 tumors in only 8% of cases, although it was associated with an upgrade in 26% of cases compared to that with TB alone [30]. In the multivariate analysis, a higher PI-RADS score was significantly associated with GG upgrade in specimens obtained using TB (*p* < 0.001) [30]. With respect to the utility of higher PI-RADS v2.0, the multivariate analysis showed that it significantly improved the predictive ability of MRI testing for biopsy GG upgrade and the C-index of the predictive nomogram (*p* = 0.001 and *p* < 0.05, respectively) [26]. A PI-RADS v2.0 score is an independent predictor of postoperative GG upgrading and should be considered when offering treatment options to men with localized PCa [26]. In our previous study, we reviewed 151 of 1268 patients with PCa undergoing RP alone at four institutions, who had only one positive PBx core, a preoperative PSA of <10 ng/mL, a biopsy GG of ≤3, and a clinical stage of T1c/T2a/T2b [28]. All enrolled cases had at least eight cores (interquartile range 10–12) taken at the time of PBx and were pathologically diagnosed with PCa [28]. The pathological diagnosis of the surgical specimens showed that 59.6% of all patients had tumors in both lobes, extracapsular invasion, or seminal vesicle invasion [16]. In addition, 21.2% of the cases were GG ≥ 4, and 29.1% of the cases were upgraded compared to the PBx specimens [28]. Although the median tumor volume was not very large (0.88 mL) in all patients, clinically insignificant PCa was diagnosed in only 8% of the cases [28]. In this study, the PI-RADS score with the highest rate of GG upgrade was 4, which was observed in 35.2% of patients. However, no statistically significant difference was found in the proportion of GG upgrades according to different PI-RADS scores. In this study, we also observed patients with downgraded GG based on pTB compared to TB. It is important to note that because PCa is a heterogeneous tumor, it may not always exhibit uniform malignancy and may show a variety of GGs depending on the location of the tumor. In addition, we examined the cancer length in the TB and pTB specimens and found that the cancer length in the pTB specimens was significantly shorter than in the TB specimens. Certainly, the addition of pTB would increase the possibility of avoiding missing PCa. However, the relatively small number of specimens obtained with pTB suggests that the GG of pTB may not accurately reflect the status of PCa, and therefore, it is necessary to be very careful when interpreting the results of pTB.

There are several limitations to this study. First, it should be noted that this is a retrospective study conducted at a single center and enrolled a relatively small number of patients, which inherently has the potential for bias. Second, most of the enrolled patients in this study had PI-RADS scores of 3 and 4, with only 13 having a PI-RADS score of 5. Therefore, the small number of potentially high-grade cases might have affected the statistical analysis of the results. Third, we defined the site of pTB as the area within 10 mm of the ROI; however, the distance from the edge of the ROI varies from case to case, and the 10 mm range may be biased. Fourth, the true size of PCa may not always be accurately assessed because of the limited number of biopsy specimens obtained from individual patients. Fifth, nine cases in which multiple ROIs were present on MRI were excluded from this study. When PI-RADS score 3 and 2 lesions were mixed in the same patient, TB and pTB were performed only on the PI-RADS score 3 lesion. In addition, some cases with large ROIs were excluded from this study because only systematic biopsies were performed. Therefore, it seemed possible that selection bias might have affected the results of this study. Finally, the primary endpoint included patients with PCa who tested negative according to PBx. Particularly, the PI-RADS score 3 group included 38 negative cases, which might have affected the statistical significance of the results. Based on these limitations, a careful interpretation of the results obtained in this study may be necessary.

## 5. Conclusions

To our knowledge, this is the first study to investigate the usefulness of pTB stratified by PI-RADS scores. In particular, we found that in patients with a PI-RADS score ≥ 4, PCa was more likely to have progressed beyond its detection using MRI. In addition, approximately 30% of the specimens obtained using pTB were upgraded compared with those obtained using TB. Therefore, our results suggested that pTB might be performed in patients with a PI-RADS ≥ 4 ROI for assessing PCa malignancy. Furthermore, the results suggested that TB and pTB could be useful in avoiding missed PCa cases and in accurately assessing GG.

## Figures and Tables

**Figure 1 diagnostics-13-02608-f001:**
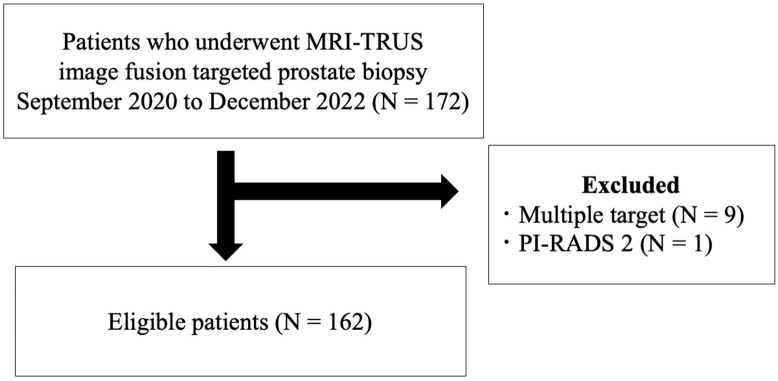
Eligibility criteria for selection of patients in this study. MRI-TRUS = magnetic resonance imaging-transrectal ultrasound; PI-RADS = Prostate Imaging Reporting and Data System.

**Figure 2 diagnostics-13-02608-f002:**
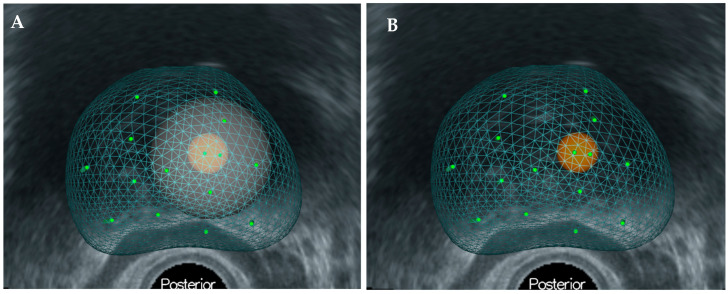
(**A**) Three-dimensional-mapping of prostate using KOELIS Trinity^®^. (**B**) The perilesional target is defined as the site within 10 mm of the region of interest.

**Figure 3 diagnostics-13-02608-f003:**
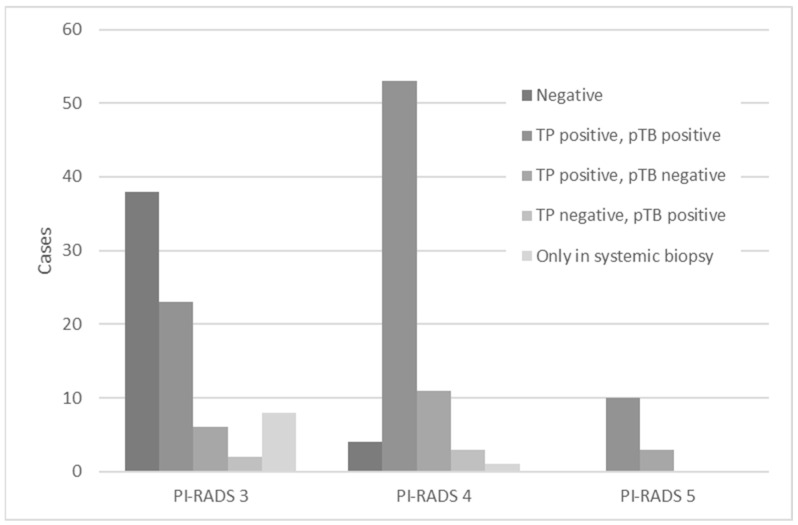
Regarding the PI-RADS, the combination of positive and negative results for TB and pTB, respectively, and the positive results for systemic biopsy, are shown separately. PI-RADS = Prostate Imaging Reporting and Data System; TB = targeted biopsy; pTB = perilesional targeted biopsy.

**Figure 4 diagnostics-13-02608-f004:**
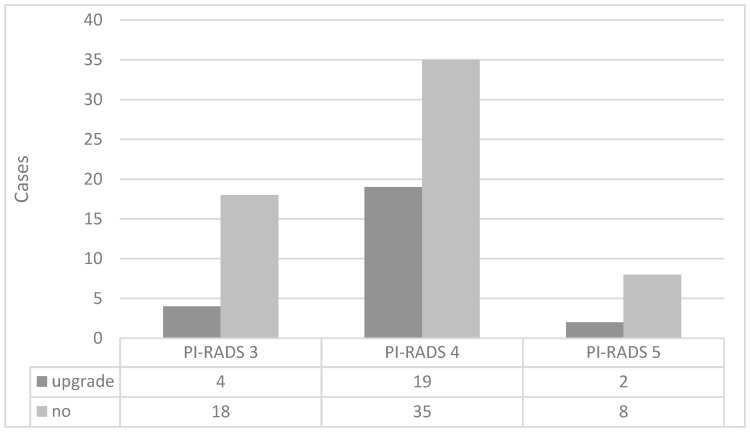
Distribution of grade group upgrades by PI-RADS score based on targeted and peripheral targeted biopsies. PI-RADS = Prostate Imaging Reporting and Data System.

**Table 1 diagnostics-13-02608-t001:** Patient characteristics.

	PI-RADS 3	PI-RADS 4	PI-RADS 5	*p*
Number of patients (%)	77 (47.5)	72 (44.4)	13 (8.0)	
Age (years, median, interquartile range)	70 (65–75.5)	71.5 (67–76.75)	72 (70–76)	0.326
PSA (ng/mL, median, interquartile range)	7.44 (4.97–10.39)	6.56 (5.19–9.67)	6.35 (5.34–16.15)	0.689
PV (mL, median, interquartile range)	45.8 (30.55–64.2)	32.75 (27.25–44.125)	35 (26.8–48)	0.0027
Biopsy core (median, min., max.)	14 (12, 18)	14 (11, 16)	14 (11, 16)	0.065
Targeted biopsy core (median, min., max.)	2 (2, 4)	2 (2, 4)	2 (2, 4)	0.650
Perilesional target core (median, min., max.)	2 (2, 6)	2 (1, 6)	2 (2, 3)	0.258
Systematic biopsy core (median, min., max.)	10 (5, 12)	10 (5, 12)	10 (7, 12)	0.148
Number of csPC cases (%)	31 (40.3)	61 (84.7)	10 (76.9)	<0.0001
Number pf cisPC cases (%)	8 (10.4)	8 (11.1)	3 (23.1)	0.413
Number of negative cases (%)	38 (49.3)	3 (4.2)	0 (0)	<0.0001

PI-RADS = Prostate Imaging Reporting and Data System; PSA = prostate-specific antigen; PV = prostate volume; csPCa = clinically significant prostate cancer; cisPCa = clinically insignificant prostate cancer; TB = targeted biopsy; pTB = perilesional targeted biopsy.

## Data Availability

Data is contained within the article.

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
