# Peer review of "Perilesional Targeted Biopsy Combined with MRI-TRUS Image Fusion-Guided Targeted Prostate Biopsy: An Analysis According to PI-RADS Scores"

_diagnostics, 2023, doi:10.3390/diagnostics13152608_

Round 1

Reviewer 1 Report

1) General comments

Tomioka et al. analyzed prostate-targeted biopsy (TB) specimens from 162 patients with prostate lesions of PI-RADS score more than 3, and investigated benefits of additional perilesional TB (pTB) for the precise diagnosis of clinically significant prostate cancer (csPCa).  This is one of the hot topics, and aspect of the present study is interesting, however, there are some flaws to be improved in the manuscript as listed below.

2) Specific comments

a) Major

1. Regarding “primary endpoint”, the detailed numbers (breakdown) of “the other” cases in Figure 3, for example, cases with positive histological diagnoses only in systemic biopsy cores, cases with positive diagnosis in TB but not in pTB, and cases with positive diagnosis in pTB but not in TB, should be described in the manuscript and Figure. This information may better clarify the direct effect of adding pTB cores for the detection rate of csPCa.

2. Regarding “secondary endpoint”, please indicate the number of cases which showed down-grade of Gleason Grade (GG) in pTB cores compared to that in TB cores. Additionally, mean and distribution of cancer length (or percentage) in each TB and pTB cores should be included in the manuscript. When only less than 1 mm (5%) of Gleason pattern 4 cancer is in a core, this core will be scored as GG4 (Gleason score 4+4=8). This factor would severely affect GG grading especially in pTB cores where a relatively small portion of cancer cells are expected to be included. Please add Discussion about this point,

3. As indicated in the first paragraph of Discussion, cases with multiple PCa and ROI in MRI are frequent in the previous studies. In the present study, only 9 cases were excluded due to multiple targetable sites for TB. How is the explanation of this issue?

4. Names of a radiologist and a urologist who performed PIRADs scoring should be described in Methods. Were there any discrepancies of MRI interpretation between these two doctors? How did the authors reach consensus in this situation? Additionally, a pathologist who reviewed the histological diagnosis should be included in the authors (if not) and indicated in the manuscript.   

b) Minor

1. Please add manufacturer information of KOELIS trinity system.

2. Line 240: “Case of both TB and pTB” means “The percentage of cases with cancer positivity in both TB and pTB”?

3. Lines 249-250: the “higher” PI-RADS score?

4. Lines 253-254: the “higher” PI-RADS v2.0 score?  

Minor editing of English language required

Author Response

July 14, 2023

Dr. Editor

The Diagnostics

Dear Editor:

Thank you very much for the review of our manuscript titled “Perilesional targeted biopsy combined with MRI-TRUS image fusion targeted prostate biopsy: Analysis according to the PI-RADS score.”

We sincerely appreciate all valuable comments and suggestions, which helped us to improve the quality of our manuscript. Our responses to the Reviewers’ comments are described below in a point-to-point manner. Appropriate changes, suggested by the Reviewers, have been introduced to the manuscript (track-changes mode in the red color font). Let me emphasize our full readiness to make any further improvements to the manuscript.

We hope that our manuscript will be acceptable for publication in the Diagnostics.

We look forward to hearing from you.

Yours sincerely,

Takuya Koie

Department of Urology

Gifu University Graduate School of Medicine

1-1 Yanagido, Gifu, Gifu 501-1194, Japan

TEL.: +81-582-30-6338

FAX: +81-582-30-6341

Responses to the reviewer's comments

We would like to thank the reviewers for taking the time and effort necessary to review the manuscript. We sincerely appreciate all the valuable comments and suggestions, which helped us to improve the quality of the manuscript.

Responses to Reviewer 1

The authors appreciate the academic reviewer’s comments. The authors’ point-by-point responses to the comments are given below.

  1. a) Major
  2. Regarding “primary endpoint”, the detailed numbers (breakdown) of “the other” cases in Figure 3, for example, cases with positive histological diagnoses only in systemic biopsy cores, cases with positive diagnosis in TB but not in pTB, and cases with positive diagnosis in pTB but not in TB, should be described in the manuscript and Figure. This information may better clarify the direct effect of adding pTB cores for the detection rate of csPCa.

Response:
The authors have added the following sentence on line 130:

We defined "the others" as patients who were negative for both TB and pTB, those who were negative for TB but positive for pTB, and those who were positive for TB but negative for pTB.

The authors have added the following sentence on line 139

“The Others” category encompasses cases that are either negative for both TB and pTB, or positive for one but negative for the other.

  1. Regarding “secondary endpoint”, please indicate the number of cases which showed down-grade of Gleason Grade (GG) in pTB cores compared to that in TB cores. Additionally, mean and distribution of cancer length (or percentage) in each TB and pTB cores should be included in the manuscript. When only less than 1 mm (5%) of Gleason pattern 4 cancer is in a core, this core will be scored as GG4 (Gleason score 4+4=8). This factor would severely affect GG grading especially in pTB cores where a relatively small portion of cancer cells are expected to be included. Please add Discussion about this point,

Response:
The authors have added the following sentences on line 151:

On the other hand, among the patients classified as "no," there were 11 cases with PI-RADS 3 whose GG matched between TB and pTB, while 7 cases showed downgrading in pTB compared to TB. In PI-RADS 4, 13 of 35 patients had downgraded GG in pTB compared to TB, while none of the patients with PI-RADS 5 had downgraded GG in pTB.

The authors have added the following sentence on line 125:

With regard to the length of cancer in prostate biopsy specimens, the median length of cancer in TB was 7 mm (interquartile range [IQR], 4-9 mm) and 5 mm (IQR, 2-7 mm) in pTB, resulting in significantly longer cancer length in TB (p = 0.021)

The authors have added the following sentence on line 294:

In this study, we also observed patients with downgraded GG of pTB compared to TB. It is important to note that because PCa is a heterogeneous tumor, it may not always exhibit uniform malignancy and may show a variety of GGs depending on the location of the tumor. In addition, we examined the cancer length in TB and pTB and found that the cancer length in pTB was significantly shorter than in TB. Certainly, the addition of pTB would increase the possibility of avoiding missing PCa. However, the relatively small number of specimens obtained with pTB suggests that the GG of pTB may not accurately reflect the status of PCa, and therefore, it is necessary to be very careful in interpreting the results of pTB.

  1. As indicated in the first paragraph of Discussion, cases with multiple PCa and ROI in MRI are frequent in the previous studies. In the present study, only 9 cases were excluded due to multiple targetable sites for TB. How is the explanation of this issue?

Response:
The authors have added the following sentence on line 312:

Fifth, nine cases in which multiple ROIs were present on MRI were excluded from this study. When PI-RADS 3 and 2 lesions were mixed in the same patient, TB and pTB were performed only on the PI-RADS 3 lesion. In addition, some cases with large ROIs were excluded from this study because only systematic biopsies were performed. Therefore, it seemed possible that selection bias might have affected the results of this study.

  1. Names of a radiologist and a urologist who performed PIRADs scoring should be described in Methods. Were there any discrepancies of MRI interpretation between these two doctors? How did the authors reach consensus in this situation? Additionally, a pathologist who reviewed the histological diagnosis should be included in the authors (if not) and indicated in the manuscript.   

Response:
The names of the three physicians were added to the authors.

The authors have added the following sentence on line 80:

by one radiologist (H.W.) and one urologist (Mas.To.) each with more than 10 years of clinical experience based on the PI-RADS v2.1 criteria. The PI-RADS score was determined by discussion between the two physicians mentioned above to make the final decision.

The authors have added the following part on line 89:

An independent pathologist (N.A.) at Chuno Kosei Hospital

The authors have added the following part on line 328:

and N.A. and H.W.

The authors have added the following part on line 330:

: funding acquisition, K.T.

The authors have added the following sentence on line 332:

Dr. Kunihiro Tsuchiya provided support for the submission fee.

  1. b) Minor
  2. Please add manufacturer information of KOELIS trinity system.

 Response:

The authors have added the following part on line 110:

(Koelis, Grenoble, France).

  1. Line 240: “Case of both TB and pTB” means “The percentage of cases with cancer positivity in both TB and pTB”?

 Response:
The authors have revised the following part on line 259:

The percentage of cases with cancer positivity in Cases of both TB and pTB

  1. Lines 249-250: the “higher” PI-RADS score?

 Response:
The authors have revised the following part on line 269:

the higher PI-RADS score

  1. Lines 253-254: the “higher” PI-RADS v2.0 score?  

Response:

The authors have revised the following part on line 271:

the utility of higher PI-RADS v2.0,

Reviewer 2 Report

Perilesional targeted biopsy combined with MRI-TRUS conducting research in the manuscript has good clinical significance and prospects, and is worthy of further promotion in clinical practice. The scientific nature of this manuscript is appropriate, and it is recommended to publish it.

Author Response

July 14, 2023

Dr. Editor

The Diagnostics

Dear Editor:

Thank you very much for the review of our manuscript titled “Perilesional targeted biopsy combined with MRI-TRUS image fusion targeted prostate biopsy: Analysis according to the PI-RADS score.”

We sincerely appreciate all valuable comments and suggestions, which helped us to improve the quality of our manuscript. Our responses to the Reviewers’ comments are described below in a point-to-point manner. Appropriate changes, suggested by the Reviewers, have been introduced to the manuscript (track-changes mode in the red color font). Let me emphasize our full readiness to make any further improvements to the manuscript.

We hope that our manuscript will be acceptable for publication in the Diagnostics.

We look forward to hearing from you.

Yours sincerely,

Takuya Koie

Department of Urology

Gifu University Graduate School of Medicine

1-1 Yanagido, Gifu, Gifu 501-1194, Japan

TEL.: +81-582-30-6338

FAX: +81-582-30-6341

Responses to the reviewer's comments

We would like to thank the reviewers for taking the time and effort necessary to review the manuscript. We sincerely appreciate all the valuable comments and suggestions, which helped us to improve the quality of the manuscript.

Response to Reviewer #2

Perilesional targeted biopsy combined with MRI-TRUS conducting research in the manuscript has good clinical significance and prospects, and is worthy of further promotion in clinical practice. The scientific nature of this manuscript is appropriate, and it is recommended to publish it.

Response:

The authors appreciate your valuable comments. The authors will continue our research so that the authors can further develop this study.

Round 2

Reviewer 1 Report

The manuscript has been improved except for the reviewer comment 1. The authors did not address important points raised during the review process.

Regarding comment 1, “detailed numbers” of each component of “the other” cases should be described in the manuscript and “Figure 3” (i.e. xx cases with positive histological diagnoses only in systemic biopsy cores, xx cases with positive diagnosis in TB but not in pTB, and xx cases with positive diagnosis in pTB but not in TB). Again, this information may better clarify the direct effect of adding pTB cores for the detection rate of csPCa.

Author Response

July 18, 2023

Dr. Editor

The Diagnostics

Dear Editor:

Thank you very much for the review of our manuscript titled “Perilesional targeted biopsy combined with MRI-TRUS image fusion targeted prostate biopsy: Analysis according to the PI-RADS score.”

We sincerely appreciate all valuable comments and suggestions, which helped us to improve the quality of our manuscript. Our responses to the Reviewers’ comments are described below in a point-to-point manner. Appropriate changes, suggested by the Reviewers, have been introduced to the manuscript (track-changes mode in the red color font). Let me emphasize our full readiness to make any further improvements to the manuscript.

We hope that our manuscript will be acceptable for publication in the Diagnostics.

We look forward to hearing from you.

Yours sincerely,

Takuya Koie

Department of Urology

Gifu University Graduate School of Medicine

1-1 Yanagido, Gifu, Gifu 501-1194, Japan

TEL.: +81-582-30-6338

FAX: +81-582-30-6341

Responses to the reviewer's comments

We would like to thank the reviewers for taking the time and effort necessary to review the manuscript. We sincerely appreciate all the valuable comments and suggestions, which helped us to improve the quality of the manuscript.

Responses to Reviewer 1

The authors appreciate the academic reviewer’s comments. The authors’ point-by-point responses to the comments are given below.

The manuscript has been improved except for the reviewer comment 1. The authors did not address important points raised during the review process. 

Regarding comment 1, “detailed numbers” of each component of “the other” cases should be described in the manuscript and “Figure 3” (i.e. xx cases with positive histological diagnoses only in systemic biopsy cores, xx cases with positive diagnosis in TB but not in pTB, and xx cases with positive diagnosis in pTB but not in TB). Again, this information may better clarify the direct effect of adding pTB cores for the detection rate of csPCa.

Response:

The authors have revised the following sentences on line 132:

Figure 3 shows the distribution of patients in the PI-RADS score 3, 4, and 5 groups, classified according to TB and pTB, and systemic biopsy test results who tested positive for both TB and pTB in the groups of PI-RADS scores of 3, 4, and 5, as well as their counterparts. Histologically, 89 cases were diagnosed positive for both TB and pTB, 9 were diagnosed positive only for systematic biopsy, 20 were diagnosed positive for TB and negative for pTB, and 5 were diagnosed negative for TB and positive for pTB. Focusing on the group that was positive for both TB and pTB, the detection rate was significantly higher in the PI-RADS 4 and 5 groups than in the PI-RADS 3 group, at 75.0% and 76.9%, respectively, compared to 28.6% in the PI-RADS 3 group In the PI-RADS 3 group, 28.6% of patients had PCa detected in both TB and pTB, whereas in the PI-RADS 4 and 5 groups, the detection rates were 75.0% and 76.9%, respectively, which were significantly higher than those in the PI-RADS 3 group (p < 0.001 and p < 0.001, respectively).

The authors have revised the following sentences on line 145:

Regarding PI-RADS, the combination of positive and negative results for TB and pTB, respectively, and the positive results for systemic biopsy, are shown separately. The proportion of patients who tested positive for both targeted and perilesional targeted biopsy of the prostate.

The authors have revised Figure 3.

Round 3

Reviewer 1 Report

The authors have substantially and satisfactory revised the manuscript according to the reviewer's comments. 

Author Response

Thank you for your kind review.